# HOXB7 Overexpression Leads Triple-Negative Breast Cancer Cells to a Less Aggressive Phenotype

**DOI:** 10.3390/biomedicines9050515

**Published:** 2021-05-05

**Authors:** Simone Aparecida de Bessa Garcia, Mafalda Araújo, Tiago Pereira, Renata Freitas

**Affiliations:** 1I3S—Institute for Innovation & Health Research, University of Porto, 4200-135 Porto, Portugal; simonebessa@med.up.pt (S.A.d.B.G.); mafalda.araujo.pereira@gmail.com (M.A.); tfpedrosa.pereira@gmail.com (T.P.); 2ICBAS—Institute of Biomedical Sciences Abel Salazar, University of Porto, 4050-313 Porto, Portugal

**Keywords:** breast cancer, HOXB7, cell phenotype, MDA-MB-231

## Abstract

HOX genes appear to play a role in breast cancer progression in a molecular subtype-dependent way. The altered expression of HOXB7, for example, was reported to promote breast cancer progression in specific subtypes. Here we induced HOXB7 overexpression in MDA-MB-231 cells, a cellular model of the Triple-Negative breast cancer molecular subtype, and evaluated the phenotypic changes in cell viability, morphogenesis, migration, invasion, and colony formation. During the phenotypic characterization of the HOXB7-overexpressing cells, we consistently found less aggressive behavior represented by lower cell viability, inhibition of cell migration, invasion, and attachment-independent colony formation capacities added to the more compact and organized spheroid growth in 3D cultures. We then evaluated the expression of putative downstream targets and their direct binding to HOXB7 comparing ChIP-qPCR data generated from HOXB7-overexpressing cells and controls. In the manipulated cells, we found enriched biding of HOXB7 to CTNNB1, EGFR, FGF2, CDH1, DNMT3B, TGFB2, and COMMD7. Taken together, these results highlight the plasticity of the HOXB7 function in breast cancer, according to the cellular genetic background and expression levels, and provide evidence that in Triple-Negative breast cancer cells, HOXB7 overexpression has the potential to promote less aggressive phenotypes.

## 1. Introduction

Breast cancer (BC) is a leading cause of cancer death among women worldwide [1]. In spite of the fact that a number of biomarkers are being used in standard clinical practice, and a growing body of others are being studied and tested, the great complexity and heterogeneity of BC still limit accurate diagnosis and therapy decision-making [2,3]. In this context, HOX gene expression has been proposed as a BC hallmark that is worth further investigation in order to improve prognosis and even develop novel targeted therapies [4,5]. These genes are organized in four clusters in the human genome: HOXA (7p15), HOXB (17q21.2), HOXC (12q13), and HOXD (2q31). Apart from their role as transcription factors, the encoded HOX proteins can also interact with other molecules to regulate cell or tissue-specific gene expression. Moreover, the genomic regions in which HOX genes are embedded produce numerous non-coding RNAs with important roles in gene regulation [6]. These “HOX products” have central functions during embryonic development and are also required for the maintenance of cellular homeostasis during adulthood [7].

In the breast, a structure that continues form and remodel throughout a woman’s life, HOX genes assume fundamental roles in normal development and disease conditions [8], as their aberrant expression is frequently associated with breast tumorigenesis [4]. This is the case of HOXB7, a gene expressed during the branching of the ductal tree and alveolar bud differentiation, and also during the involution process after lactation [4]. This HOX gene has a tendency to be overexpressed in primary BC with a more prominent overexpression in metastasis [9] and has also been shown to be overexpressed in a variety of cell lines representative of distinct molecular subtypes [4,10]. The highest overexpression of HOXB7 was detected in Luminal A and B models (MCF7 and BT474, respectively), and the lowest overexpression was observed in Triple-Negative (MDA-MB-468 and MDA-MB-231) and HER2+ models (SKBR3) [4,10]. These subtype-specific expression profiles suggest that HOXB7 may have distinct roles according to the genetic background [4,10].

In vivo and in vitro BC models have associated the increased levels of HOXB7 with the activation of TGFB signaling and with the expression levels of HER2 and EGFR [11,12]. In addition, TGFB/SMAD3 signaling is activated by HOXB7, because SMAD3 phosphorylation seems to be higher in primary BC from Hoxb7/Her2 double-transgenic mice than from Her2/neu single-transgenic mice. These studies provide evidence that HOXB7 may bind to the TGFB2 promoter in double-transgenic tumors, in which TGFB2 expression is frequently higher than in single-transgenic mice. Moreover, TGFB2 expression was found to be higher in two BC cell lines transfected with HOXB7 (SKBR3, MDA-MB-231) and the knockdown of TGFB2 in MDA-MB-231 cells overexpressing HOXB7 seems to inhibit lung metastasis in mice.

While some HOXB7 targets have been uncovered in Luminal and HER2+ molecular subtypes, little is known about the molecular pathways affected by HOXB7 overexpression in Triple-Negative BC (TNBC), which have cells lacking Estrogen Receptor (ER), Progesterone Receptor (PR), and HER2 expression. This BC molecular sub-type represents up to 15% of the cases and has the worse overall survival in every stage and sub-stage when compared to non-TNBC [13,14]. This high aggressiveness is represented by early relapses, few and non-targeted treatment options, and low response durability [15]. Using the MDA-MB-231 line, a cellular model of TNBC, we generated an HOXB7-overexpressing clone in order to analyze the associated phenotypes and uncover the targets affected by HOXB7 modulation in this genetic background. We found that in an HOXB7-overexpressing clone: cell viability was lower; cell migration, invasion, and colony formation in soft agar were inhibited; and sphere formation in 3D cultures appeared more organized. In addition, we detected that, along with decreased CTNNB1 expression, the HOXB7-overexpressing clone presented enriched binding of HOXB7 to the promoter regions of EGFR, DNMT3B, COMMD7, and CDH1.

Overall, our results suggest that particular levels of HOXB7 can inhibit the aggressive behavior of TNBC cells possibly through its direct or indirect interaction with genes such as CTNNB1, FGF2, CDH1, EGFR, DNMT3B, and COMMD7. This adds important information to the current understanding of the HOXB7-dependent signaling pathways in TNBC cells for which the discovery of reliable predictive biomarkers is imperative to improve patient prognosis and treatment.

## 2. Materials and Methods

### 2.1. Cell Culture

Human BC cells MDA-MB-231 (MDA231) were grown in Dulbecco’s modified Eagle’s medium, DMEM 1X (Gibco, Paisley, UK) supplemented with 10% (*v*/*v*) heat-inactivated fetal bovine serum (FBS, Biowest, Riverside, MO, USA), and 1X antibiotic solution penicillin-streptomycin, pen-strep (Gibco, Grand Island, NE, USA), and maintained in 5% CO_2_ at 37°C. Transfected cells were grown in a medium with 700 µg/mL of geneticin (Gibco, Bangkok, Thailand).

### 2.2. Stable Transfection Assay

MDA231 cells were transfected with constructs pCMV6-AC-GFP-HOXB7 (Origene, RG204495) or pCMV6-AC-GFP (Origene, PS100010), using TurboFectin 8.0 (Origene). Three different cell transfectants were obtained and named as Empty Vector (EV), for the cells transfected with the control Empty Vector (pCMV6-AC-GFP); B7, for the pool of cells transfected with pCMV6-AC-GFP-HOXB7 vector and overexpressing variable levels of HOXB7; D3, for the clone obtained from sorted cells transfected with pCMV6-AC-GFP-HOXB7 vector.

### 2.3. RNA Expression Analyses

Total RNA was extracted from cells using TRIzol™ (Ambion, Carlsbad, CA, USA) and reverse transcribed to cDNA using the High-Capacity cDNA Reverse Transcription Kit (Applied Biosystems, Vilnius, Lithuania). Quantitative PCR (qPCR) was implemented with iTaq™ Universal SYBR Green Supermix (Bio-Rad, Hercules, CA, USA) and amplification conditions were: 95 °C for 3 min (min), 40 cycles of 95 °C for 10 s (s) and 60 °C for 30 s. followed by the default dissociation curve capture. Expression of genes was normalized to that of glyceraldehyde-3-phosphate dehydrogenase (GAPDH) and relative quantification analyses was performed using Schmittgen and Livak method [16]. Primer sequences are indicated in Appendix A.

### 2.4. ChIP-qPCR Assay

DNA/protein crosslink was achieved by incubating the samples in 1% formaldehyde solution under agitation. ChIP-qPCR reactions were performed using 2 μL of the purified sample and the run conditions were: 95 °C for 3 min, 40 cycles of 95 °C for 15 s, 55 °C for 30 s and 72 °C for 30 s. followed by the default dissociation curve capture. The analyses were made using the “percent input” method according to Lacazette [17]. Primer sequences are described in Appendix A.

### 2.5. Protein Expression Assays

Total protein extractions were obtained using lysis solution and cytoplasmic and nuclear protein fraction extractions were obtained using ab113474 Nuclear Extraction Kit (Abcam). Protein lysates were separated on a 12% SDS-PAGE gel prepared with 40% Acrylamide–Bisacrylamide 29:1 (Invitrogen, Carlsbad, CA, USA) and 4X separating buffer (Alfa Aesar, Ward Hill, MA, USA) for the separating gel or 4X stacking buffer (Alfa Aesar, Karlsruhe, Germany) for the stacking gel. After run the proteins were blotted onto a nitrocellulose membrane of the iBlot^®^ gel transfer stacks (Kiryat Shmona, Israel) using the iBlot™ dry transfer system (Life Technologies, Herzliya, Israel). The primary and secondary antibodies used are listed in Appendix A. The assessment of the bands’ density was made in the ImageLab software (BioRad) using the measures of Tubulin (for total and cytoplasmic protein fractions) and HDAC1/Lamin B1 (for nuclear protein fraction) as loading controls. A representative Western blot of the purity of nuclear and cytoplasmic protein fractions extraction is shown in Appendix A.

### 2.6. MTT Assay with Docetaxel Treatments

For the MTT assay [3-(4,5-dimethylthiazol-2yl)-2,5-diphenyltetrazolium bromide] (EMD Millipore, Shanghai, China), cells were seeded in 96-well plates (1 × 10^4^ cells/well) and maintained in 5% CO_2_ at 37 °C in complete media supplemented with 5% FBS for 24 h. At this time point (24 h), the Docetaxel (Sigma-Aldrich, Shanghai, China) treatments with 5 nM and 50 nM started. The cell viability measurements were made by absorbance read at 570 nm using the Synergy™ 2 Plate reader (BioTek) at 24 h, 48 h, 72 h, and 96 h post-seed. The control cells were treated with ETOH in the volume corresponding to the biggest drug concentration used. An additional assay was made for the measurement of cell viability in a standard medium without ETOH.

### 2.7. On-Top 3D Cell Culture

MDA231 wild-type (WT), EV, and D3 cells were dissociated to obtain a suspension containing single cells. The cells (2 × 10^3^), suspended in 400 µL of completed media containing 3% of Matrigel^®^ Matrix Basement Membrane growth factor reduced (Corning, Bedford, MA, USA), were seeded in 8-well glass chamber slides containing a pre-prepared bed of 30 μL Matrigel^®^ Matrix Basement Membrane growth factor reduced and incubated in 5% CO2 at 37 °C for 8 days. The cells’ spheroids morphologies were analyzed and registered every two days in Axiovert 200 M inverted fluorescent microscope (Carl Zeiss). On day 4, 100 µL of media containing 3% Matrigel^®^ Matrix Basement Membrane growth factor reduced were added to the wells to prevent the effects of medium evaporation and nutrient scarcity.

### 2.8. Wound-Healing Assay

Cells were seeded in 24-well plates at a density of 2 × 10^5^ cells/well in 500 µL of Dulbecco’s modified Eagle’s medium, DMEM 1X (Gibco, Paisley, UK) supplemented with 5% (*v*/*v*) Charcoal Stripped Fetal Bovine Serum Qualified One Shot™ (Gibco, Monterrey, Mexico), for cells mitosis synchronization, and 1× antibiotic solution penicillin-streptomycin (pen-strep, Grand Island, NE, USA). After wound, cells were incubated for 15 h, and images were captured every 3 h in the InCell Analyser 2000 Automated fluorescence widefield HCS microscope. The extent of wound closure was measured using the MRI wound-healing tool from ImageJ software.

### 2.9. Invasion Assay

The invasion assay was made using the 24-well plate growth factor reduced Corning^®^ Matrigel^®^ Invasion chambers 8 μm pore size (Corning, Bedford, MA, USA) and 24-well Control Inserts 8 μm pore size (Corning) according to manufacturer’s instructions. Briefly, cells (1 × 10^5^) in 500 µL serum-free medium were plated into the upper chamber and the bottom wells were filled with 750 µL complete medium. The cells were incubated in 5% CO_2_ at 37 °C for 16 h. Then, cells in the upper chamber were removed using cotton swabs and the cells invading the bottom of the membrane were fixed with 4% paraformaldehyde for 20 min. The nuclei were stained with DAPI (Sigma-Aldrich, Darmstadt, Germany) 1 µg/mL plus 1% Triton X-100 in 1X PBS (Gibco, Paisley, UK) for 15 min followed by two washes in Milli-Q H2O. Ten random fields from each membrane were photographed using the Zoe fluorescent cell imager (Bio-Rad) and the cells’ nuclei were counted using the Analyze Particles tool from ImageJ software.

### 2.10. Soft Agar Colony Formation Assay

This assay was made according to Franken, Borowicz, and colleagues [18,19]. The cells (1.5 × 10^4^/well) were seeded in 6-well plates and incubated for four weeks in a 5% CO_2_ at 37 °C. The colonies, without staining, of nine random fields, were counted using the Zoe fluorescent cell imager (Bio-Rad). MCF7 cells cultured in the same medium as MDA231 cell line were used as a parameter for Luminal behavior analyses.

### 2.11. Slow Aggregation Assay

This assay was performed according to Boterberg and colleagues [20]. Briefly, 2 × 10^4^ cells per well were seeded over an agar layer in a 96-well plate with and without the addition of MB2, an anti-CDH1 antibody (kindly provided by Dr. Joana Paredes), as a way to show if the formed aggregates are dependent on the CDH1/CTNNB1 complex functionality. The cells were photographed after 24 h, 48 h, 72 h, and 96 h in an Axiovert 200 M inverted fluorescent microscope (Carl Zeiss). The MDA231-WT and MCF7 cells were used, respectively, as negative and positive controls of the aggregation capacity. MDA231-WT cells do not express CDH1. Therefore, they do not aggregate while MCF7 cells, which express high levels of CDH1, form well-defined aggregates when CDH1/CTNNB1 complex is active.

### 2.12. Statistical Analyses

The statistical differences were determined by unpaired *T*-test with Welch’s correction or by Brown–Forsythe and Welch ANOVA tests using the Prism8 software (GraphPad Software, La Jolla, CA, USA). *p*-values were considered statistically significant when *p* ≤ 0.05. Data were reported as the mean ± SD of three or more independent experiments.

## 3. Results

### 3.1. HOXB7 Overexpression in MDA231 Cells

HOXB7 is typically overexpressed in BC cell lines in comparison with normal cells [4,10]. However, the level of HOXB7 overexpression is lower in MDA231 cells (Triple-Negative, claudin-low) in comparison to MCF7 (Luminal A), BT474 (Luminal B), and MDA468 (Triple-negative, basal-like) [4,10]. We selected the MDA231 cells to conduct functional assays aiming to explore the mechanistic effect of HOXB7 upregulation in TNBC. These cells were transfected with pCMV6-AC-GFP plasmid vector as control generating the Empty Vector (EV) cells and with pCMV6-AC-GFP-HOXB7 plasmid vector generating the B7 and D3 cells. While the B7 cells have heterogeneous levels of HOXB7 overexpression, the D3 cells result from a single clone and, therefore, have homogenous levels of HOXB7 overexpression. After transfection, GFP expression was detected in EV, B7, and D3 cells, indicating that the exogenous *HOXB7* have been translated (Appendix A).

Subsequent mRNA expression analysis revealed that the levels of HOXB7 were identical in EV and WT cells, suggesting that the transfection procedure does not affect HOXB7 expression (Figure 1A). In contrast, the B7 and D3 cells presented higher levels of HOXB7 expression (*p* = 0.02 and *p* = 0.013, respectively) when compared to EV cells. This seems to be corroborated by the levels of GFP expression in the HOXB7-manipulated cells versus the controls (Appendix A). In addition, the HOXB7 expression was significantly higher in D3 than in B7 cells (*p* = 0.015) (Figure 1A). In the nuclear fraction (Figure 1C), HOXB7 expression was significantly higher in D3 cells in comparison with EV (*p* = 0.026) (Figure 1B). B7 cells presented higher levels of the HOXB7 protein with a borderline significance in comparison with the control (*p* = 0.049) and no differences were detected in the cytoplasmic fraction between WT, EV, B7, and D3 cells. In summary, we confirmed that pCMV6-AC-GFP-HOXB7 transfection led to HOXB7 overexpression in B7 and D3 cells, and we further detected that only D3 cells present significant HOXB7 overexpression at the nuclear protein level, which is expected due to their characteristic function as transcription factors.

### 3.2. Phenotypic Characteristics of HOXB7-Overexpressing Cells

#### 3.2.1. Compact Spheroid Organizations in 3D Culture

The three-dimensional (3D) culture on a reconstituted basement membrane results in the formation of spheroids that recapitulate several aspects of glandular architecture in vivo [21]. The morphology of D3 cell growth in the conventional 2D culture did not show impacting differences in comparison to the morphology observed in WT and EV cells (Appendix A). However, we hypothesized that 3D cellular organization could be different in the HOXB7 overexpressing cells. To address this question, the cells were grown on top of Matrigel^®^ for 8 days (Figure 2A), which allowed us to see that D3 cells form smaller spheroids with a more compact growth without the formation of membrane protrusions. In contrast, WT and EV cells showed a more spread out growth with numerous protrusion formations. Interestingly, the spheroids’ growth pattern of D3 cells resembles that observed for MCF7 cells (Luminal A) [22].

#### 3.2.2. Lower Cell Viability and No Effect on Sensitivity to Docetaxel Treatments

The effect of HOXB7 overexpression on cell viability and Docetaxel sensitivity was analyzed in EV and D3 cells by MTT assay at different time points (Figure 2B). At 48 h and 96 h post-plating, the D3 cells exhibited lower viability in comparison to EV cells (*p* = 0.002 and *p* = 0.0001, respectively) (Figure 2A). We then evaluated the sensitivity to Docetaxel, a chemotherapeutic drug that induces different types of cell death depending on the administrated concentration [23]. We observed that, in the presence of Docetaxel (5 nM and 50 nM), the cells do not reach the exponential growth as observed in absence of the drug (Figure 2B). The viability differences, observed at 24 h post-plating (time 0 h for drug treatments) among the cells before and after Docetaxel treatment, are probably related to cell loading and/or cell attachment issues. Despite the differences in 24 h, it is possible to observe the lower viability of the D3 cells kept with ETOH in 96 h compared to EV cells (*p* = 0.005) corroborating the findings in Figure 2A. In the ETOH assay, the cells were sub-confluent at 96 h, possibly due to the differences observed at 24 h. In conclusion, no differences in response to Docetaxel treatments were observed over time.

#### 3.2.3. Lower Migration and Invasion Capacities

The cell migration capacity is essential for disease progression and is considered the first step for lymphovascular invasion and tumor metastasis [24]. The data obtained by the wound-healing assay showed that D3 cells had a significant delay in wound closure capacity, nearly 30%, in all analyzed time points when compared to EV cells (Figure 3; *p* = 0.027 at 3 h and *p* < 0.0001 at 6 h, 9 h, 12 h, and 15 h). The invasion capacity, which is another important skill developed by cancer cells to successfully spread both locally and more distantly, was analyzed by the Matrigel^®^ invasion chamber assay. It was observed that D3 cells present a lower capacity (*p* = 0.004) to invade the Matrigel^®^ layer in comparison to EV cells with an invasion index of 0.4 (Figure 4), which means that the invasion capacity of D3 cells is approximately 60% lower than in EV cells.

#### 3.2.4. Lower Soft Agar Colony Formation Efficiency

The ability of the transformed cells to grow and form colonies independently of a solid surface is a hallmark of carcinogenesis [19]. We observed that the D3 cells formed significantly fewer colonies than WT (*p* = 0.041) and EV cells (*p* = 0.007) (Figure 5). Moreover, when compared to a Luminal A cell line (MCF7), the MDA231 WT and EV cells showed a higher number of colonies (*p* = 0.009 and *p* = 0.007, respectively), while D3 cells showed a similar colony number as MCF7 cells. In summary, D3 cells presented a colony formation efficiency that represents about 50% of the efficiencies observed in WT and EV cells, which is similar to what is observed in MCF7 cells.

### 3.3. Putative Downstream Targets of HOXB7 in TNBC Cells

#### 3.3.1. Impact on CTNNB1 Expression

The current knowledge on the direct and/or indirect targets of HOXB7 in BC is scarce; therefore, the search for these molecules is crucial to better understand HOXB7 functions in distinct molecular subtypes. Given the results obtained in the 3D cultures, showing a more compact and organized spheroid formation in D3 cells, we speculate that molecules involved in cell–cell adhesion and cytoskeletal organization (i.e., CDH1/CTNNB1 complex) could be influenced by HOXB7 overexpression. Following this line of thought, we performed a slow aggregation assay in order to verify if the phenotypes mentioned above could be related to changes in CDH1 functionality, given that MDA231 cells do not express CDH1 and consequently do not form aggregates in this assay. However, no alterations were observed in the D3 cell aggregation profiles, in the presence and absence of the MB2 antibody, in comparison to WT and EV cells (Appendix A) and therefore, the phenotypes observed in 3D cell culture do not seem directly linked to variations in CDH1 activity. Next, we analyzed the expression of CTNNB1, a multi-functional molecule with key roles in normal and disease conditions [20]. Interestingly, we found significant downregulation of CTNNB1 mRNA (*p* = 0.021, Figure 6A) and total protein in D3 cells (*p* = 0.007, Figure 6B,C).

#### 3.3.2. Interaction with CDH1, FGF2, and CTNNB1 Promoters and, When Overexpressed, EGFR, DNMT3B, and COMMD7 Genes Become New Targets

Based on previous studies, several putative targets of HOXB7 may explain its mechanistic role when overexpressed in BC cells: EGFR, FGF2, CTNNB1, CDH1, DNMT3B, and COMMD7. HOXB7 seems to establish direct interactions with EGFR in MCF7 [12] and BT474 cells [25] and with FGF2 in BT474 cells [25]. In addition, the knockdown of CTNNB1 in TNBC cell lines (MDA231 and HCC38) significantly impairs their ability to migrate and form anchorage-independent colonies [26] besides showing a downregulation in our HOXB7-overexpressing cells (Figure 5B,C). Moreover, reduced expression of CDH1 is linked to the invasion capacity of cancer cells [27]. However, its interaction with HOXB7 has not been explored in BC cells. Regarding DNMT3B, its function relates to de novo methylation, which has an important impact on epigenetic regulation of several genes [28] and TNBC prognosis [29]. It was also shown that HOXB7 binds directly to the TGFB2 promoter in MCF7 cells and upregulates this same molecule in MDA231 cells leading to increased cell migration and invasion17. Finally, the analysis of ChIP-Seq data [25] evaluating HOXB7 binding to the whole genome in BT474, suggests that COMMD7 might be a direct target of HOXB7 in these cells (Appendix A). This gene was already suggested to be involved in the progression of hepatocellular [30] and pancreatic carcinomas [31] but no references exist regarding BC. We analyzed COMMD7 expression in a panel of BC cells and found upregulation in BT474 and MDA231 (Appendix A). We then used ChIP-qPCR to explore the interaction of HOXB7 with all putative targets mentioned above. We detected three HOXB7 interaction patterns (Figure 6): (1) interaction in EV and D3 cells with no increments in D3 cells for CTNNB1 and FGF2; (2) interaction only in D3 cells for EGFR, DNMT3B, and COMMD7; and (3) interaction in EV and D3 cells with an increment in D3 cells for CDH1. No interactions between HOXB7 and TGFB2 were found in EV or D3 cells.

## 4. Discussion

Our analyses of the HOXB7 basal mRNA expression profile [32] suggest that, among BC cell lines representing the distinct BC molecular subtypes, HOXB7 is heterogeneously expressed according to the genetic background of the cells, which may lead to variable functions. In fact, the lowest levels of overexpression were observed in SKBR3 (HER2+) and MDA231 cells (TNBC, claudin-low), which represent the molecular subtypes with worse prognosis. In contrast, the highest expression levels were observed in the luminal A and B cells (MCF7 and BT474), representing molecular subtypes with better prognosis [32,33]. Surprisingly, our data reveal that the induced overexpression of HOXB7 in MDA231 cells generates less aggressive phenotypes, represented by smaller and more compact spheroid formation in 3D culture, and lower cell viability, migration, invasion, and anchorage-independent colony formation. This contradicts the idea that HOXB7 is always an oncogene in BC [4]. Moreover, the search for potential downstream targets showed that the HOXB7-overexpressing cells, along with decreased CTNNB1 expression, presented enriched direct interaction of HOXB7 with the promoter regions of EGFR, DNMT3B, COMMD7, and CDH1 (Figure 7).

The maintenance of breast cells on a reconstituted basement membrane results in a 3D-growth that recapitulates several aspects of glandular architecture in vivo that are lost in 2D culture, as demonstrated with MCF10A cells [21,34]. Interestingly, MDA231 HOXB7-overexpressing cells (D3) formed smaller and more compact spheroids compared to the spread and branched growth shown by the control cells. When cultured in 2D conditions, the EV and D3 cells did not show impactful differences in their morphology. Moreover, the spheroids of MDA231 HOXB7-overexpressing cells resembled those generated by BT474 and MCF7 cells [35,36], reinforcing the idea that HOXB7 upregulation relates to a more “luminal-like” phenotype.

The plasma membrane protrusions are the result of the continuous synthesis and remodeling of the cytoskeleton actin filaments and are closely related to the promotion and driving of cell migration. In 3D cell movement, these protrusions are collectively known as invadossomes, which establish close contact with the Extracellular Matrix (ECM) and perform a proteolytic matrix degradation to invade the connective tissues [37]. Thus, the decreased protrusion formations observed in the 3D cultures could explain the impaired capacity of HOXB7-overexpressing cells to migrate in a 2D culture and to invade through the Matrigel^®^ matrix, as will be discussed in further detail.

The impact of HOXB7 overexpression on BC cell proliferation has been demonstrated in different cellular models. Caré and colleagues [38] showed that HOXB7 overexpression in SKBR3 cells (ER−/PR−/HER2+) leads to increased cell proliferation through FGF2 upregulation. Ma and colleagues [39] corroborated these results in MCF7 cells (ER+/PR+/HER2−), in which HOXB7 downregulation decreased the proliferation ratios. However, an in vivo approach [40] showed that HOXB7 overexpression alone is insufficient to induce tumor formation and had a dual role when co-overexpressed with HER2. The Hoxb7/Her2 transgenic mice, compared to Her2 transgenic animals, showed a delayed tumor onset reflected in a decreased tumor multiplicity but, once the tumor was established, Hoxb7 promoted tumor progression leading to the formation of larger masses and to a higher index of pulmonary micro-metastasis. Additionally, in silico analyses of public microarray data showed that a high level of HOXB7 predicts a poor outcome in HER2+, but not in HER2- patients [40]. This is a strong indication that the HOXB7 role in tumor progression is dependent on the cellular genetic background, especially concerning the HER2 profile and ECM interactions.

The cell/cell and cell/ECM interactions are important for cellular architectural maintenance and growth control, among other processes [41]. The MDA231 cell line was established from cells recovered from a patient’s pleural effusion [42]. Therefore, these cells had already accomplished the entire metastatization process and, additionally, maintained its metastatic capability in vivo. Our results show that HOXB7 overexpression leads MDA231 cells to a decreased capacity to form colonies in an anchorage-independent environment. Moreover, HOXB7 overexpressing cells showed a colony formation capacity similar to that observed in MCF-7 cells, which is also a line established from a pleural effusion, but without metastatic potential in vivo.

Akekawatchai and colleagues [43], through proteomic analyses, identified 54 proteins expressed only in suspended and adherent MDA231, but not in MCF7 cells in the same conditions. Could MDA231 HOXB7-overexpression have changed the expression profiles of some proteins acquiring a genetic background closer to the MCF7 cells? Could these proteins be related to the other phenotypic changes observed? These questions deserve attention. Controversially, the HOXB7 overexpression in SKBR3 cells (HER2+) increased their ability to form colonies in a semisolid medium [38]. Once more, the contradictory findings may relate to the cells’ genetic profiles and, consequently, to the pathways modulated in each particular condition. Thus, characterization of the HOXB7 targets is still an urgent issue for a better understanding of the HOXB7 functions in distinct cellular contexts. Here we provided evidence for the physical interaction established between HOXB7 and the promoter regions of EGFR, DNMT3B, CDH1, CTNNB1, FGF2, TGFB2, and COMMD7 in MDA231 cells. In addition, we also found particular interactions enriched after overexpressing HOXB7 in these cells, which will be useful for the search of biomarkers for TNBC prognosis and treatment (Figure 7).

The Epidermal Growth Factor Receptor (EGFR) overexpression is recognized as a driver mechanism in the initiation, progression, and therapy resistance of several carcinomas such as lung, breast, and pancreatic cancers [44], added the interaction HOXB7/EGFR promoter region already demonstrated [25]. EGFR is also an important target for multiple chemotherapeutic regimens [45]. Jin and colleagues [12] showed that tamoxifen resistance in MCF7 cells is related to a progressive increasing in HOXB7 expression levels, along with the upregulation of EGFR and its ligands. Although we do not observe EGFR expression differences between EV and D3 cells (Appendix A), by making a parallel between the genetic backgrounds of MDA231 and MCF7 cells concerning HOXB7, EGFR, and ER expression [32,46,47], we can infer that MDA231 is HOXB7-low, EGFR-high, and ER-negative, while MCF7 is HOXB7-high, EGFR-low, and ER-positive. Therefore, in both cells, we observed an inverse correlation between the three molecules. Additionally, BT474 cells are HOXB7-high, EGFR-intermediary, ER-positive, but resistant to hormone therapy. Adding to our finding that HOXB7/EGFR interaction occurred only in HOXB7-overexpressing MDA231 cells (HOXB7-high, EGFR-high, ER-negative), we have more evidence that HOXB7 action is dependent, not only on its own expression but also on the molecules that are active in the cell, named ER, EGFR, and HER2. Thus, different genetic profile combinations could lead to different responses for the same stimulus in the same tissue.

Similarly, we only found an interaction between HOXB7/DNMT3B in HOXB7-overexpressing cells. No studies exist correlating HOXB and DNMT3B molecules. However, it is widely known that the levels of DNMTs, (DNMT3B, DNMT3A, and DNMT3L) are often increased in various cancer tissues and cell lines (Appendix A), and may account for the hypermethylation of tumor suppressor genes in a variety of malignancies [48]. As the HOXB7 methylation, which has already been described in several cancers and has been shown to relate with patient’s prognosis [49], the search for genes that are hypermethylated in response to HOXB7-binding to DNMT3B is also an important and interesting approach to pursue.

CDH1 was another gene shown to interact with HOXB7 in MDA231 cells with an interaction increment in which HOXB7 overexpression was induced. It has already been demonstrated that HOXB7-overexpression in MCF10A cells causes a reduction in CDH1 expression [9]. In BC, CDH1 impaired function potentiates the metastatization and relates with worse prognosis and shorter overall survival of patients [27]. Despite the higher HOXB7/CDH1 interaction found in MDA231 HOXB7-overexpressing cells, the CDH1 functionality did not change between EV and D3 cells, as accessed by the aggregation assay. However, CDH1 interacts with the actin cytoskeleton [27]. Therefore, the increment of the HOXB7/CDH1 interaction in MDA231 HOXB7-overexpressing cells could be related to the absence of protrusion formations in these cells. Additionally, CDH1 works in complex with CTNNB1 to ensure cell-cell adhesion between epithelial cells [20]. The balance between CDH1 and CTNNB1 expression is responsible for the cell adherens-junction maintenance or for the CTNNB1 release from the complex followed by its degradation or nuclear translocation, where it forms complexes with members of the TCF/LEF family to activate the transcription of target genes involved in self-renewal, Epithelial to Mesenchymal Transition (EMT) and cell proliferation [50,51]. Moreover, in BC patients, nuclear and cytosolic accumulation of CTNNB1, but not the membrane-associated form, is associated with reduced overall survival [52].

Surprisingly, the CTNNB1 promoter region showed to be an HOXB7 target in MDA231 EV and D3 cells, adding the fact that the CTNNB1 mRNA and protein levels were decreased in HOXB7-overexpressing cells. Xu et al. [26] demonstrated that CTNNB1-knockdown impaired the ability of the MDA231 and HCC38 cells to migrate. Moreover, the HCC38 cells showed a decreased capacity to form anchorage-independent colonies in soft agar, a lower stemness potential, a reduced tumorigenic potential in vivo, and increased sensitivity to chemotherapeutic agents (cisplatin and doxorubicin). Another study [53] showed that CTNNB1 downregulation effectively reverses EMT in the MDA231 and BT549 TNBC cells and suppresses the metastatic potential of MDA231 cells in vivo. These findings may relate to the phenotypes of migration and invasion observed in D3 cells. Concerning the stemness, we analyzed the mRNA expression of the c-Myc, NANOG, OCT4, and SOX2 transcription factors, known as master regulators of pluripotency and stemness [54], and did not observe any differences between EV and D3 cells (Appendix A). This could be due to the fact that the signals released by CTNNB1 knockdown to promote changes in stemness are not the same as those provided by the HOXB7 overexpression effects on CTNNB1 expression. The same explanation may apply to the absence of sensitivity changes to docetaxel treatments. Chemotherapeutic drugs have different action modes and mechanisms of resistance. Cisplatin and doxorubicin are DNA-interfering drugs, while docetaxel acts on tubulin impairing the dynamics of the microtubules and consequently, inhibiting cell cycle progression [55,56].

Another gene that showed a direct interaction with HOXB7 in both EV and D3 cells was FGF2. The HOXB7/FGF2 interaction has already been described in melanoma [57] and BT474 cells [25]. We found that FGF2 is expressed only in MCF10A (normal breast) and MDA468 cells (TNBC, basal) (Appendix A). Even in MCF7, BT474, and D3 cells, which show high levels of HOXB7, the FGF2 expression was undetectable (Appendix A). Despite the existence of HOXB7/FGF2 interaction in different cell models and the described relationship between the increased expression of FGF2 and BC better prognosis [58], further studies are required to establish in which context HOXB7 may be involved in FGF2 expression regulation.

Although the direct binding of HOXB7 to the TGFB2 promoter has already been described in MCF7 cells [59], this interaction was not observed in MDA231 EV and D3 cells. Liu and colleagues [59] showed that TGFB2 knockdown in MDA231 cells overexpressing HOXB7 leads to a decrease in the migration and invasion capacities, along with a dramatic inhibition of lung metastasis formation in vivo. The TGFB2 mRNA levels were also analyzed in EV and D3 cells but no differences were detected (Appendix A). The role of TGFB2 in BC progression is ambiguous, as it was shown to display tumor-suppressing and -enhancing effects [60]. Thus, a number of questions still remain in order to completely understand the regulation of TGFB2 and its interaction with pathways activated in the course of tumor progression.

Finally, COMMD7, a gene that we found to be overexpressed in BT474 and MDA231 lines (Appendix A) and that potentially presents enriched interactions with HOXB7 in BT474 [25], was found to be a target of HOXB7 in D3 cells. The COMMD family is a group of proteins that act as scaffold proteins to facilitate the assembling of crucial molecules involved in the control of several biological processes such as the NF-κB signaling [61]. No information exists on the specific role of COMMD7 in BC but data exist revealing that COMMD7 expression levels are upregulated in hepatocellular carcinoma tissues [61] and pancreatic ductal adenocarcinoma (PDAC) tissues and cell lines [37,38]. Interestingly, HOXB7 expression is also high in liver and pancreatic cancers [4]. Therefore, it is possible to infer that HOXB7 and COMMD7 may have interaction-dependent functions as well in BC. A summary of the main findings of this work is presented in the scheme shown in Figure 7.

Taken together, our results highlight the plasticity of the HOXB7 protein and its relevance as a transcription factor in BC, a function that is highly conditioned by the cellular genetic background and its own expression levels. In addition, we provide evidence that in Triple-Negative breast cancer cells, HOXB7 overexpression has the potential to promote less aggressive phenotypes.

## 5. Conclusions

Triple-Negative breast cancer cells expressing high levels of HOXB7 mRNA and nuclear protein show a less aggressive phenotype represented by lower viability and decreased migration, invasion, and anchorage-independent colony formation capacities along with a more compact spheroid formation in 3D culture. In MDA231 cells overexpressing HOXB7, the HOXB7 protein interacts with CTNNB1, FGF2, CDH1, EGFR, DNMT3B, and COMMD7 promoter regions. Moreover, these cells present CTNNB1 downregulation, which relates to the phenotypes observed. Thus, the present study raises important questions concerning, not only the downstream targets of HOXB7 but also the molecules that could be involved in HOXB7 regulation and nuclear translocation, as well as the genes that are differentially methylated in response to HOXB7/DNMT3B interaction. In addition, our results suggest that the HOXB7 role in breast cancer is strictly dependent on the cellular genetic background, especially regarding ER, EGFR, and HER2 expression, along with the modulation of the levels of this transcription factor.

## Figures and Tables

**Figure 1 biomedicines-09-00515-f001:**
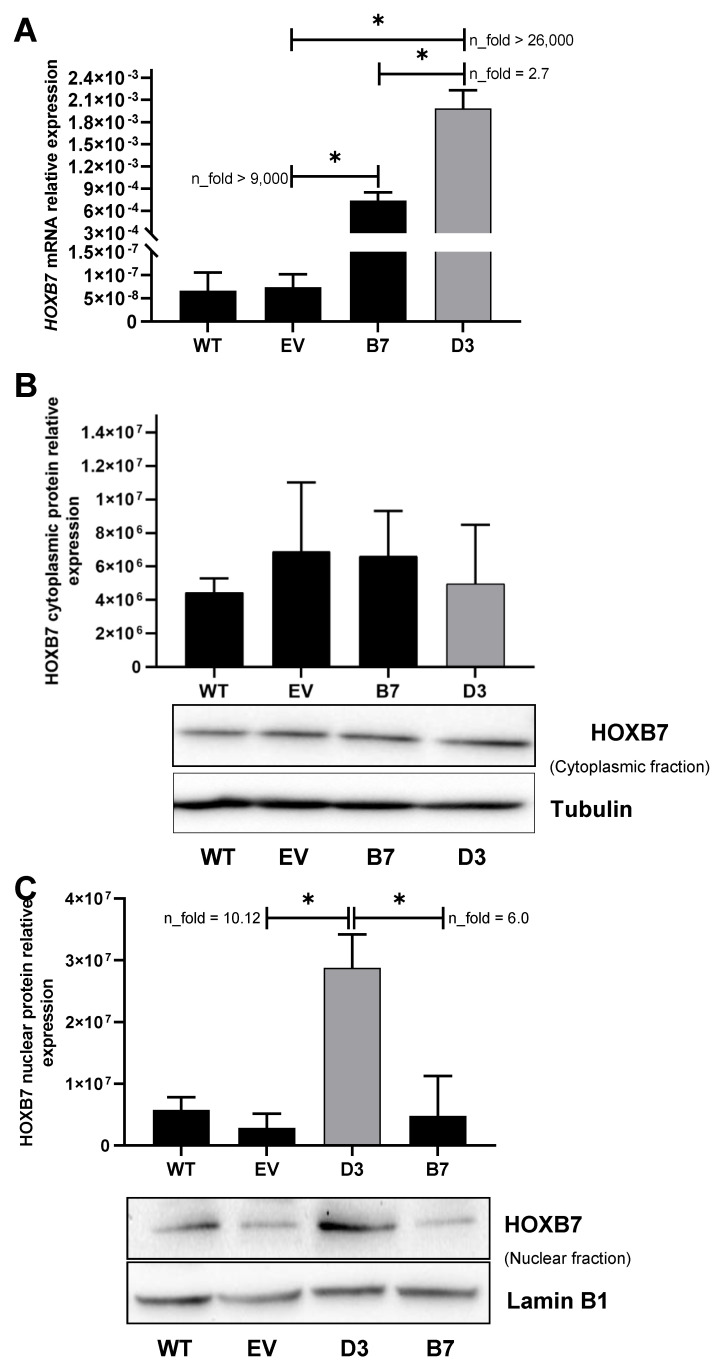
HOXB7 expression in MDA231 cells after transfection. (**A**) HOXB7 mRNA expression in wild-type cells (WT), empty-vector transfected cells (EV), B7 (pool of cells transfected with HOXB7-expression vector), and D3 (clone of cells transfected with HOXB7-expression vector). *p*-Values for each comparison were: EV×B7 (*p* = 0.0199), EV×D3 (*p* = 0.0127), and B7×D3 (*p* = 0.0146). (**B**) HOXB7 cytoplasmic protein expression in WT, EV, B7, and D3 cells. (**C**) HOXB7 nuclear protein expression in WT, EV, B7, and D3 cells. *p*-Values for each comparison were: EVxD3 (*p* = 0.0263) and B7×D3 (*p* = 0.00486). The bars represent the mean ± SD obtained in three independent experiments. *, Statistically significant differences, *p* ≤ 0.05, obtained by Brown–Forsythe and Welch ANOVA tests (Multiple comparisons) with Games–Howell’s correction. n-fold represents the fold change in the indicated comparison taking EV or B7 values as reference.

**Figure 2 biomedicines-09-00515-f002:**
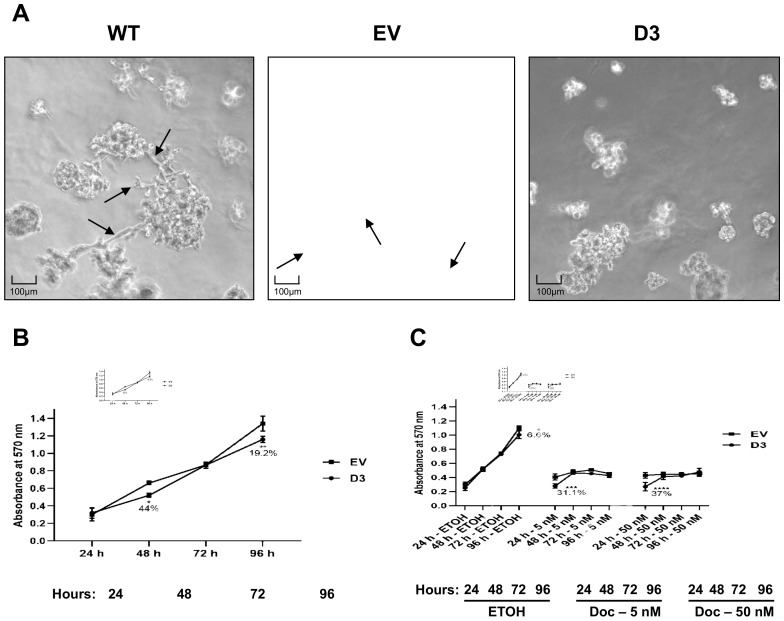
HOXB7 overexpression effect on cell spheroids in 3D cultures and on viability and Docetaxel sensitivity in MDA231 cells. (**A**) Cell spheroids in 3D cultures from WT, EV, and D3 cells. WT and EV cells have similar patterns of 3D-growth with spread spheroids with protrusions (arrows). D3 cells grow in smaller and more compact spheroids without protrusion formations. (**B**) MTT cell-viability measure in EV and D3 cells at 24 h, 48 h, 72 h, and 96 h post-plating. *p*-values for 48 h and 96 h were *p* = 0.0140 and *p* = 0.0017, respectively. (**C**) EV and D3 cell sensitivity to 5 nM and 50 nM Docetaxel treatment at 24 h, 48 h, 72 h, and 96 h post-plating. *p*-values for ETOH-96 h was *p* = 0.0102; for Docetaxel 5 nM-24 h was *p* = 0.0003 and for Docetaxel 50 nM-24 h was *p* < 0.0001. Cells were kept in ETOH as control of vehicle action. The graphs represent the mean ± SD obtained in four independent experiments. *, Statistically significant differences (*, *p* ≤ 0.05, **, *p* ≤ 0.01, ***, *p* ≤ 0.001 and ****, *p* ≤ 0.0001) obtained through 2way ANOVA test (Multiple comparisons) with Sidak’s correction. The % values refer to the difference in D3 cell behavior in comparison to EV cells.

**Figure 3 biomedicines-09-00515-f003:**
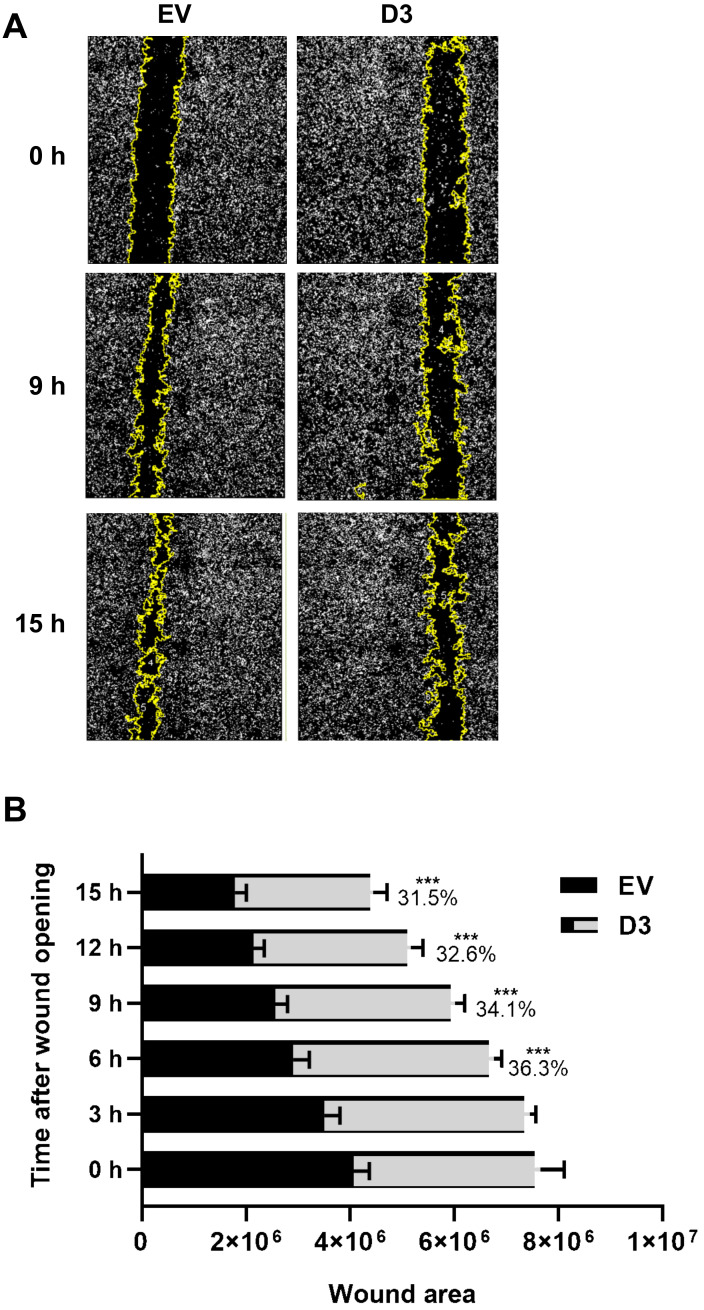
HOXB7 overexpression effect on MDA231 cells migration capacity. (**A**) Representative images of the wound-healing assay in EV and D3 cells at 0 h, 9 h, and 15 h after the scratch wound. Images captured using InCell Analyser 2000, 10x objective. (**B**) Wound areas of the EV and D3 cells along the time. The graph represents the mean ± SD obtained in four independent experiments. ***, Statistically significant differences, *p* ≤ 0.001, obtained through 2way ANOVA test (Multiple comparisons) with Sidak’s correction. The *p*-values for the D3 × EV comparisons are *p* = 0.0001 at 6 h and *p* = 0.0003 at 9 h, 12 h, and 15 h. The percentage values refer to the difference in D3 cells wound areas in comparison to EV cells.

**Figure 4 biomedicines-09-00515-f004:**
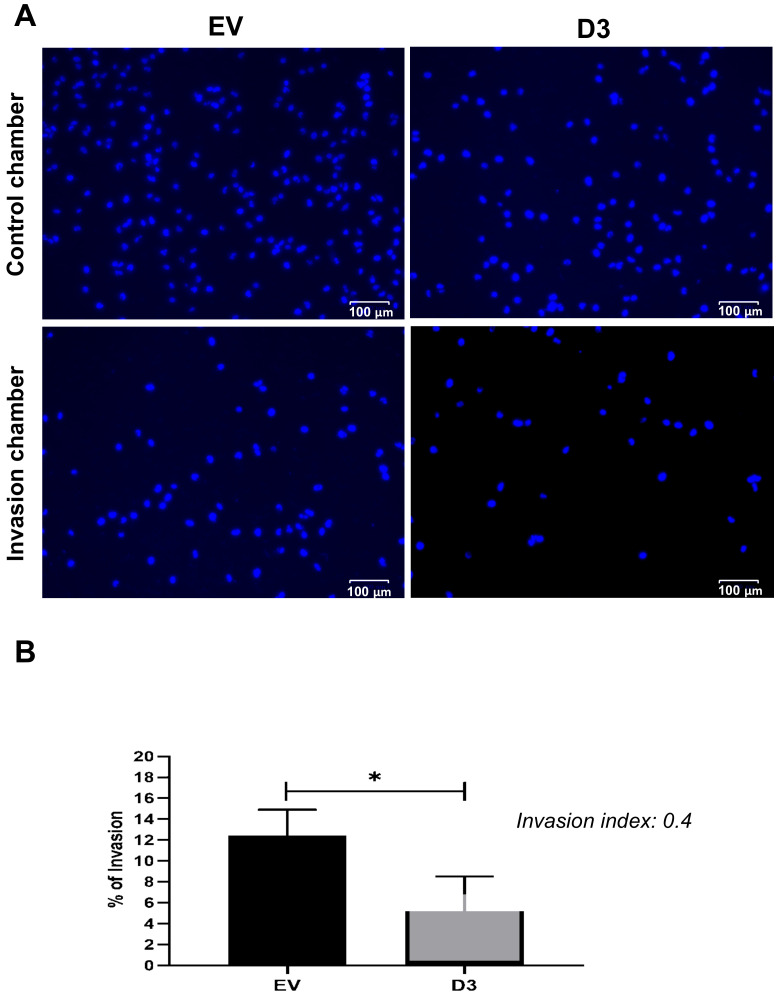
HOXB7 overexpression effect on MDA231 cell invasion capacity. (**A**) Representative images of the nuclei of the EV and D3 cells that invaded the Matrigel^®^ layer of the invasion chamber and migrate through the control chambers. (**B**) Invasion percentage of EV and D3 cells with the respective invasion index. The graph represents the mean ± SD obtained in three independent experiments. * Statistically significant differences, *p* ≤ 0.05, obtained by unpaired *T*-test with Welch’s correction (*p* = 0.0395).

**Figure 5 biomedicines-09-00515-f005:**
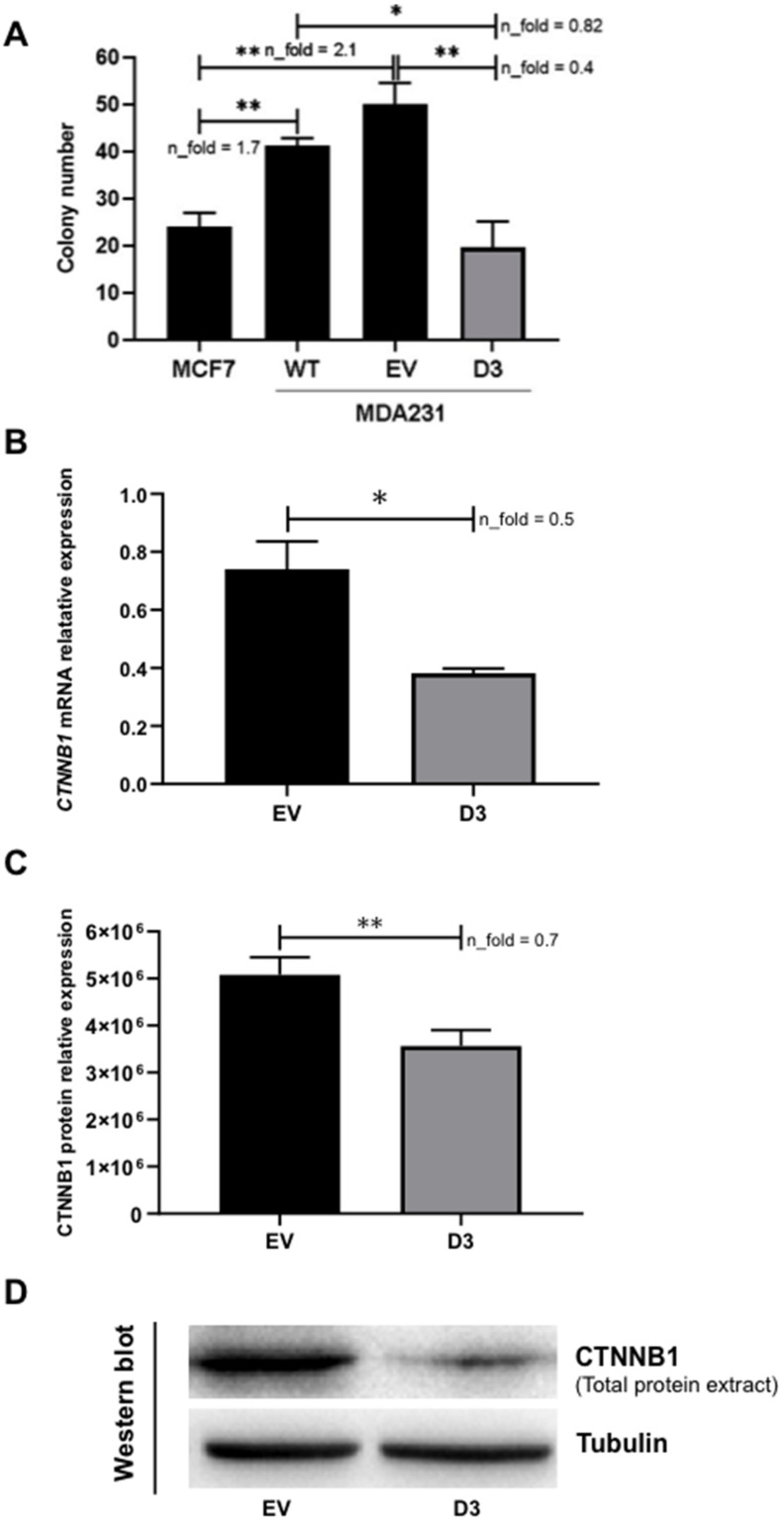
HOXB7 overexpression effect on anchorage-independent colony formation and on CTNNB1 expression in MDA231 cells. (**A**) Number of colonies formed by MCF7 and MDA231 (WT, EV, and D3). The graph represents the mean ± SD obtained in three independent experiments. *, Statistically significant differences, *p* ≤ 0.05, obtained by Brown–Forsythe and Welch ANOVA tests with Games–Howell’s correction. *p*-values were: MCF7xWT (*p* = 0.0093), MCF7xEV (*p* = 0.0071), WTxD3 (*p* = 0.0408), and EVxD3 (*p* = 0.0071). n_fold represents the fold change in the indicated comparison and taking MCF7 or EV values as reference. (**B**,**C**) CTNNB1 mRNA and total protein relative expression in EV and D3 cells. (**D**) Western blot representative of the CTNNB1 total protein analyses in EV and D3 cells with the expression of Tubulin as the loading control. The bars represent the mean ± SD obtained in three independent experiments. *, Statistically significant differences (*, *p* ≤ 0.05 and **, *p* ≤ 0.01), obtained by unpaired *T*-test with Welch’s correction. *p*-values for mRNA and protein expression analyses were *p* = 0.0207 and *p* = 0.0066, respectively. n_fold represents the fold change in the expression of D3 cells in comparison to EV cells.

**Figure 6 biomedicines-09-00515-f006:**
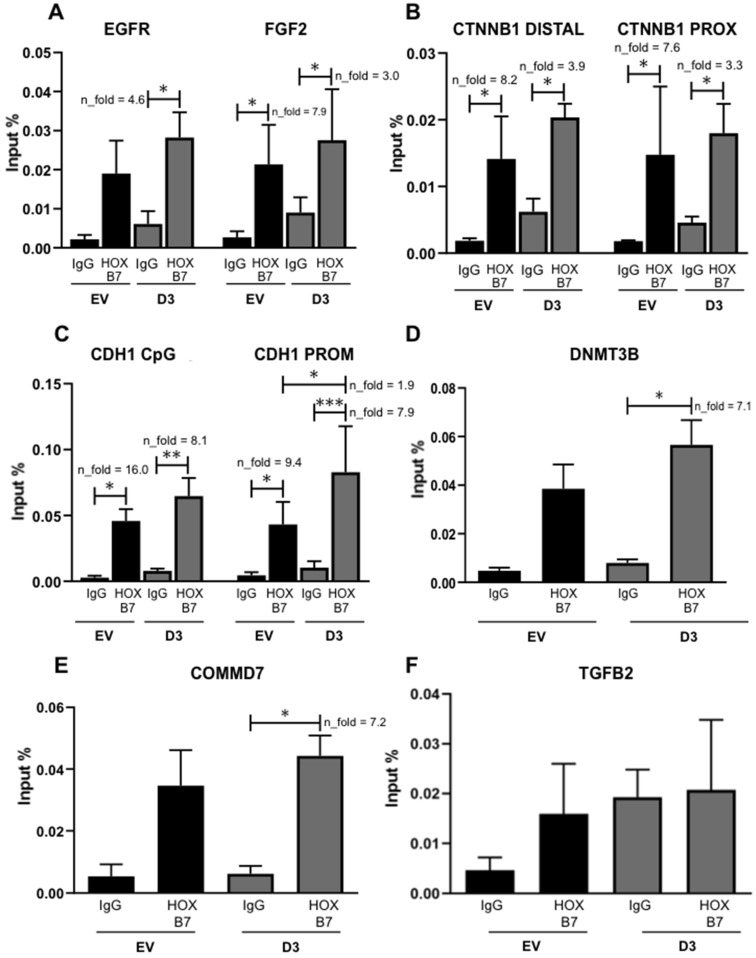
HOXB7 protein direct interaction with the promoter regions of EGFR (**A**), FGF2 (**A**), CTNNB1 (**B**), CDH1 (**C**), DNMT3B (**D**), TGFB2 (**E**), and COMMD7 (**F**). IgG, Immunoprecipitations with anti-IgG antibody (interaction negative control). HOXB7, Immunoprecipitations with anti-HOXB7 antibody. The bars represent the mean ± SD obtained in three independent experiments. *, Statistically significant differences, (*, *p* ≤ 0.05, **, *p* ≤ 0.01 and *** *p* ≤ 0.001) obtained by 2way ANOVA test (Multiple comparisons) with Bonferroni’s correction (EGFR, FGF2, CTNNB1 DISTAL, CTNNB1 PROX, CDH1 CpG, and CDH1 PROM) or by Brown–Forsythe and Welch ANOVA tests (Multiple comparisons) with Games–Howell’s correction (DNMT3B, TGFB2, and COMMD7). The calculated *p*-values were: EGFR (IgG_D3 x D3 = 0.01), FGF2 (IgG_EV x EV = 0.035, IgG_D3 × D3 = 0.036), CTNNB1 (DISTAL: IgG_EV × EV = 0.032, IgG_D3 × D3 = 0.011. PROX: IgG_EV × EV = 0.022, IgG_D3 × D3 = 0.017), CDH1 (CpG: IgG_EV× EV = 0.018, IgG_D3 × D3 = 0.002. PROM: IgG_EV × EV = 0.034, IgG_D3 × D3 = 0.0001, EV × D3 = 0.031), DNMT3B (IgG_D3 × D3 = 0.033), and COMMD7 (IgG_D3 × D3 = 0.013). n_fold represents the fold change in the indicated comparison and taking IgG or EV values as reference.

**Figure 7 biomedicines-09-00515-f007:**
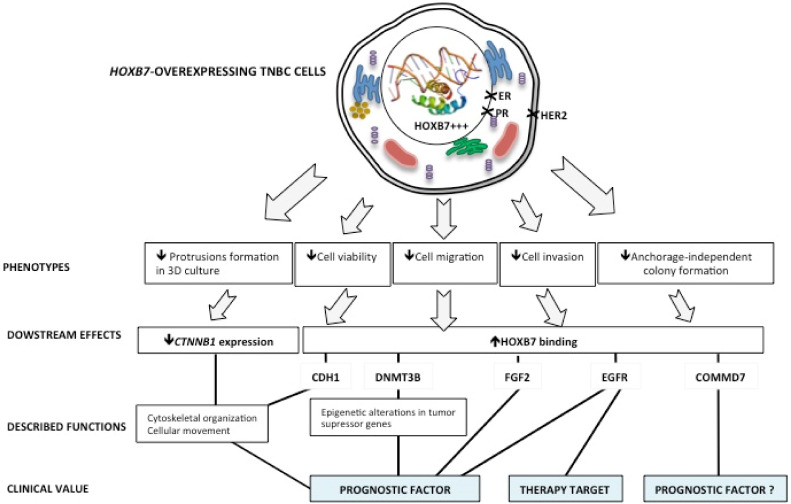
Summary of the HOXB7 nuclear overexpression effects in the TNBC cell line MDA-MB-231. The HOXB7 nuclear overexpression leads the MDA-MB-231 cells to a less aggressive phenotype represented by a more organized spheroid formation in 3D culture and reduced cell viability, migration, invasion, and anchorage-independent colony formation. The observed phenotypes could be related to changes in CTNNB1 expression and to HOXB7 binding to genes with important roles in breast cancer progression. An exception was made for COMMD7 whose function in breast cancer is still understudied. HOXB7 tridimensional structure within the nucleus is by Emw—Own work, CC BY-SA 3.0, https://commons.wikimedia.org/w/index.php?curid=8820026 (accessed on 29 April 2021).

## Data Availability

Not applicable.

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
