# Peer review of "HOXB7 Overexpression Leads Triple-Negative Breast Cancer Cells to a Less Aggressive Phenotype"

_biomedicines, 2021, doi:10.3390/biomedicines9050515_

Round 1
Reviewer 1 Report
The basic problem of this manuscript remains unsolved. pCMV6-AC-GFP-HOXB7 did not increased protein level of HOXB7 rather to change it's localization. pCMV6-AC-GFP-HOXB7 should increases both mRNA and protein levels of HOXB7 through increasing gene transcription. I think the result of HOXB7 overexpression dose not make sense. Hence, the following results were not convincible. My previous comments seem not to be considered.
Author Response
We are grateful and appreciated the straightforwardly of this reviewer. However, we are sorry that the misunderstandings of certain aspects of our article made him/her provide a negative opinion on our manuscript. We apologize if we were not clear enough and we tried to explain and revise the text in order to avoid misinterpretations from future readers. The most relevant “misunderstandings” were the following: the reviewer thinks that we claim that after HOXB7-manipulation in MDA-MB-231 the cellular localization changes. That is not what we claim, we just present the first figure to evaluate if our functional assay worked, and, to this end, we evaluated the levels of HOXB7 protein after generating the stable clones. However, the figure that we presented contains the nuclear and cytoplasmic fraction separated. This is because we realized that if we compared the entire cellular content we were not able to see differences (we show that in supplementary information). This is normal in transcription factors such as HOXB7 because their function is mainly in the nucleus. Please find below a point-by-point response to the specific comments of this reviewer.
Reviewer 2 Report
This is a totally new submission of previously refused manuscript presenting: HOXB7 overexpression leads Triple-Negative breast cancer cells to a less aggressive phenotype.
The authors have accomplished almost all remarks previously given.
Author Response
We are grateful for the positive evaluation of our work.
Reviewer 3 Report
The edited manuscript was well written and clear.
Some typo problems can be solved, such as the "P" should be in italic fonts and a space needs to add between number and unit.
Author Response
We are grateful for the positive evaluation of our work and we solved the typos and did an extra revision of the text.
Round 2
Reviewer 1 Report
Authors' response: This is normal in transcription factors such as HOXB7 because their function is mainly in the nucleus. This concept is not correct.
I still can not agree this concept.
e.g. When we transfect c-myc gene, a well-known transcription, c-myc overexpression estimated by Western blotting will not be restricted in the nucleus.
e.g. When we transfect histone H3 gene, a well-known protein to regulate chromosome remodeling in the nucleus, H3 overexpression estimated by Western blotting will not be restricted in the nucleus.
To validate Authors' hypothesis, expression of pCMV6-AC-GFP-HOXB7 must be determined by Western blotting using the anti-GFP antibody.
GFP (26 kDa)-HOXB7 (24 kDa) expresses at the higher molecular weight (~50 kDa) than wild type HOXB7 (24 kDa). It is fine that endogenous wild type HOXB7 remains the same after gene transfection. Authors must confirm that exogenous GFP-HOXB7, not endogenous HOXB7, is, indeed, overexpressed in their D3 cells
Author Response
We took into deep consideration the comment of the referee, to whom we are extremely grateful, and provide a point-by-point rebuttal below (italic for the referee's comments and bold for our responses):
- Authors' response: This is normal in transcription factors such as HOXB7 because their function is mainly in the nucleus. This concept is not correct. I still can not agree this concept. e.g. When we transfect c-myc gene, a well-known transcription, c-myc overexpression estimated by Western blotting will not be restricted in the nucleus.
R: The referee says that we consider it “normal” to find higher levels of HOXB7 in the nucleus in our HOXB7-manipulated cells because the protein in question is a transcription factor. The referee then gives the example of c-myc that appear to be not restricted to the nucleus estimated by Western blotting. We never said that our expression was restricted to the nucleus. What we said is that the most significant changes are found in the nuclear portion, there is also expression in the cytoplasm but not different from the controls. And yes, this is “normal” because the HOXB7 tends to go to the nucleus to perform its function as a transcription factor. If c-myc levels were studied in the cytoplasmic versus nuclear portion, the authors would probably find differences in the c-myc manipulated cells. In fact, the sub-cellular localization of c-myc was studied after gene silencing/overexpression and it is different in the cytoplasmic and nuclear portions (e.g. JECCR, 2018, 37:194 https://doi.org/10.1186/s13046-018-0861-9). In addition, several authors provided exclusively data from the nuclear factor, while studying c-myc (e.g. Cancer Research, 2008 68(16):6643-51; DOI:10.1158/0008-5472.CAN-08-0850). Our option was to show all data, from the cytoplasm and from the nuclear portion, because we find it important as a first indication of the gain-of-function of HOXB7 transcription factor.
- g. When we transfect histone H3 gene, a well-known protein to regulate chromosome remodeling in the nucleus, H3 overexpression estimated by Western blotting will not be restricted in the nucleus.
R: Again, we never said that HOXB7 becomes restricted to the nucleus in the transfected cells and again, researchers working with histone H3 also found it relevant to study the nuclear portion given the function of this protein (The Journal of Immunology, 2014, 193(1), DOI:10.4049/jimmunol.1302923).
- To validate Authors' hypothesis, expression of pCMV6-AC-GFP-HOXB7 must be determined by Western blotting using the anti-GFP antibody. GFP (26 kDa)-HOXB7 (24 kDa) expresses at the higher molecular weight (~50 kDa) than wild type HOXB7 (24 kDa). It is fine that endogenous wild type HOXB7 remains the same after gene transfection. Authors must confirm that exogenous GFP-HOXB7, not endogenous HOXB7, is, indeed, overexpressed in their D3 cells.”
R: The concept of using an anti-GFP antibody to access the exogenous HOXB7 expression is not correct. The antibodies are specific to their targets in a way that, independently of what is bound to GFP, the size of the observed band will be the same. To access this binding the correct assay is the immunoprecipitation. If the question is the GFP expression, we now present images showing HOXB7 higher expression in transfected cells (new Supplementary Figure 2) and we edited the text according to this new piece of information (indicated with Track-changes). In addition, given that biological effects are observed only in D3 cells in the subsequent experiments it is accurate to infer that they are the result of exogenous HOXB7 that translocated into the nucleus. The EV cells were transfected with the same vector (without HOXB7) and do not show any functional difference.
As a result of this reflection, we made this first section of the results much more succinct and “straight to the point”: to prove that the transfection was efficient and generated HOXB7 overexpressing cells. We are grateful to the reviewer for making us think and double-check the literature to make sure our data is accurate and that the nuclear extracts are frequently used alone for better “resolution” of the western blot analyses dealing with proteins acting mainly in the nucleus.
If any additional revision or clarification is needed, please let us know. We greatly appreciate your attention in this matter.
Round 3
Reviewer 1 Report
No questions
This manuscript is a resubmission of an earlier submission. The following is a list of the peer review reports and author responses from that submission.
Round 1
Reviewer 1 Report
Authors provided numerous experimental results to clarify the tumor-suppressive effect of HOXB7 on breast cancer. There are too many flaws in this study. Experimental results are not solid to support Authors' hypothesis.
- Gene overexpression should dramatically increases both HOXB7 mRNA and protein levels. Authors claimed that HOXB7 plasmid transfection affects its localization of HOXB7 protein rather than increases it's protein level. This is very weird. I can not agree this concept. Based on this flaw, I think following experimental results are acceptable.
- The quantitative results for Western blotting in Figure 1C is not consistent with the representative image.
- Figure 2: The image of 3D spheroid is very not clear. I think 2D spheroid is easier to do, and its enough to evaluate Authors' hypothesis.
- Figure 3B. This quantitative method is very confused to me. Authors just need to quantify the migratory area.
- Figure 6: There is no significant difference between EV and D3 using HOXB7 antibody. Hence, this result can not support Authors' hypothesis: HOXB7 associates with these genes promoters.
- Labeling in the supplementary figure S1 is very rough.
- Supplementary Figure S5 is very blurred.
- Supplementary S8: EV exhibits significant effect ???
- Supplementary results do not support results in the main text.
Overall, results in this study do not connect each other. Some results even fight each other (Figure 1 vs Supplementary Figure S2; Figure 5A vs Supplementary Figure S3-4.....).
English editing is also required.
Reviewer 2 Report
The paper presents a study of protein overexpression-related to the HOXB7 in triple-negative breast cancer cell line.
As the world-wide leading cancer disease among women, it is still a hot topic to study.
Notes
INTRODUCTION
I consider important to cite the following latest article dealing with BC disease contributing also to the topic
In spite of the fact that a number of biomarkers are being used in standard clinical practice, and a growing body of others are being studied and tested, the great complexity and heterogeneity of BC still limit accurate diagnosis and therapy decision-making [2, https://doi.org/10.1007/s11033-019-04598-w].
INTRODUCTION
At the end of intro, please specify more your outcome as for studied protein related to BC.
MATERIALS AND METHODS
This part is quite exhaustive what seems to make the processing long-lasting. Please provide a graphic of the experimental workflow used for the study, to make it more clear for the readers.
RESULTS
This part is well written and understandable. Please, provide a 3D structure of the HOXB7
CONCLUSION
Discuss more about the impact of your outcome to the future cure of BC and your future plans to improve the method to make it more simple.